# General principles for the formation and proliferation of a wall-free (L-form) state in bacteria

**Romain Mercier, Yoshikazu Kawai, Jeff Errington\***

Centre for Bacterial Cell Biology, Institute for Cell and Molecular Biosciences, Newcastle University, Newcastle upon Tyne, United Kingdom

**Abstract** The peptidoglycan cell wall is a defining structural feature of the bacterial kingdom. Curiously, some bacteria have the ability to switch to a wall-free or 'L-form' state. Although known for decades, the general properties of L-forms are poorly understood, largely due to the lack of systematic analysis of L-forms in the molecular biology era. Here we show that inhibition of peptidoglycan precursor synthesis promotes the generation of L-forms from both Gram-positive and Gram-negative bacteria. We show that the L-forms generated have in common a mechanism of proliferation involving membrane blebbing and tubulation, which is dependent on an altered rate of membrane synthesis. Crucially, this mode of proliferation is independent of the essential FtsZ based division machinery. Our results suggest that the L-form mode of proliferation is conserved across the bacterial kingdom, reinforcing the idea that it could have been used in primitive cells, and opening up its use in the generation of synthetic cells.

## Introduction

The peptidoglycan (PG) cell wall is a major defining feature of bacterial cells and is present in all known major bacterial phyla, suggesting that the wall was present in the last common ancestor of the whole bacterial lineage (*Errington, 2013*). PG is composed of long glycan strands cross linked by short peptide bridges, forming a meshwork that covers the whole cell. The wall has a variety of important functions, including the following: maintenance of cell shape, protection from mechanical damage, and generation of turgor by restraining the outward osmotic pressure exerted on the cytoplasmic membrane. It is the target for our best antibiotics (β-lactams, glycopeptides, etc), and fragments of the wall trigger important innate immune responses. The wall is assembled by polymerization and cross linking of a precursor molecule, termed lipid II, which is synthesized in the cytoplasm and then transferred to the cell surface for wall assembly (*Typas et al., 2012*).

Despite its importance, many bacteria, both Gram-positives and Gram-negatives, are capable of switching into a cell wall deficient state, called the 'L-form' (*Allan et al., 2009*). Generally, L-forms were generated under osmoprotective conditions (e.g. in the presence of 0.5 M sucrose) by long term and repeated passage, sometimes for years, in the presence of β-lactam antibiotics that inhibit PG synthesis (*Allan, 1991*). However, the lack of reproducible and tractable model systems prevented the development of consensus views of the common properties of L-forms derived from different bacteria.

We have recently undertaken a systematic analysis of the L-form transition in the experimentally tractable Gram-positive bacterium *Bacillus subtilis*. We have defined genetic pathways required to elicit a reproducible and rapid switch to the L-form state and identified genes required specifically for L-form growth in this organism (*Leaver et al., 2009*; *Dominguez-Cuevas et al., 2012*; *Mercier et al., 2012*, *2013*). Our analysis of *B. subtilis* L-form growth led to two unexpected findings. First, that when dividing in the L-form state, *B. subtilis* becomes completely independent of the FtsZ (tubulin) based division machinery (*Leaver et al., 2009*) and the MreB (actin) cytoskeleton (*Mercier et al., 2012*).

**\*For correspondence:** jeff. errington@newcastle.ac.uk

**Competing interests:** The authors declare that no competing interests exist.

**eLife digest** Bacterial cells are surrounded by a cell wall made of a molecule called peptidoglycan. This wall is important for many aspects of cell survival including the maintenance of cell shape and protection from mechanical damage. However, many bacteria are able to switch to a state in which they don't have a cell wall. Although this wall-free state was discovered several decades ago, little is known about its general properties because there isn't a quick and reliable method for making such bacteria.

Recently, it has been shown that bacteria of the species *Bacillus subtilis* can rapidly switch to the wall-free state when the production of peptidoglycan is reduced. Here, Mercier et al. show that the same method also works for a wide range of bacterial species.

The wall-free states of the various species share the same unusual way of dividing to produce daughter cells. Normally, bacterial cell division is a highly controlled process involving a protein called FtsZ that accumulates at the site of cell division. In bacteria without walls, on the other hand, cell division does not require FtsZ, but instead depends on the rate of production of new cell membrane. Excessive production of membrane leads to the cell changing shape, resulting in spontaneous separation into daughter cells.

The results suggest that this form of cell division is conserved across all bacteria. It is possible that this is an ancient mechanism that may have been used by the ancestors of modern bacteria, before the evolution of the cell wall. In future, this simple form of cell division could prove useful the development of synthetic living cells.

Instead, the L-forms divide by a remarkable process of cell shape deformation, including blebbing, tubulation, and vesiculation, followed by spontaneous resolution (scission) into smaller progeny cells (**Kandler and Kandler, 1954**; **Leaver et al., 2009**). We recently showed that L-form proliferation in *B. subtilis* simply depends on excess membrane synthesis, leading to an increase in the surface area to volume ratio (**Mercier et al., 2013**). Upregulation of membrane synthesis can be driven directly, by mutations affecting the regulation of fatty acid synthesis, or indirectly, by shutting down PG precursor synthesis, which presumably depends on a regulatory circuit that we do not yet understand. To complicate matters, the growth of *B. subtilis* L-forms requires a second mutational change, most commonly affecting the *ispA* gene (**Leaver et al., 2009**), which probably works by compensating for a metabolic imbalance that occurs when cells grow in the absence of wall synthesis (Kawai and Mercier, unpublished).

To date, we have restricted our attention to *B. subtilis* L-forms. In this study, we have shown that inhibition of PG precursor synthesis seems to be an efficient method to create stable L-forms from a range of diverse bacteria, including a Gram-negative *Escherichia coli*. We have also characterized several key properties of these L-forms, including their mode of proliferation, and we have found them to be strikingly reminiscent of *B. subtilis* L-forms, in the following ways: (i) mode of cell proliferation using cell shape deformation followed by a spontaneous formation of progeny cells; (ii) dispensability of the normally essential cell division machinery; and (iii) key role for the membrane synthesis rate in cell proliferation.

The strikingly similar properties of L-forms from different bacterial lineages reinforces the idea that their mode of cell proliferation could have been used in primitive bacteria before the invention of the cell wall, and that they could be used in the generation of synthetic cells.

## Results

### Inhibition of the PG precursor pathway promotes stable L-form proliferation in diverse bacteria

We previously showed that excess membrane synthesis is required for L-form proliferation and that this can be achieved directly by upregulation of the fatty acid synthase (FAS II) system or indirectly by inhibition of PG precursor synthesis (*Figure 1A*). We do not yet understand the basis for coupling of PG precursor and fatty acid synthesis but the effect on *B. subtilis* is shown in *Figure 1B*. Although inhibiting the PG precursor pathway was lethal on non-osmoprotective nutrient agar (NA) plates

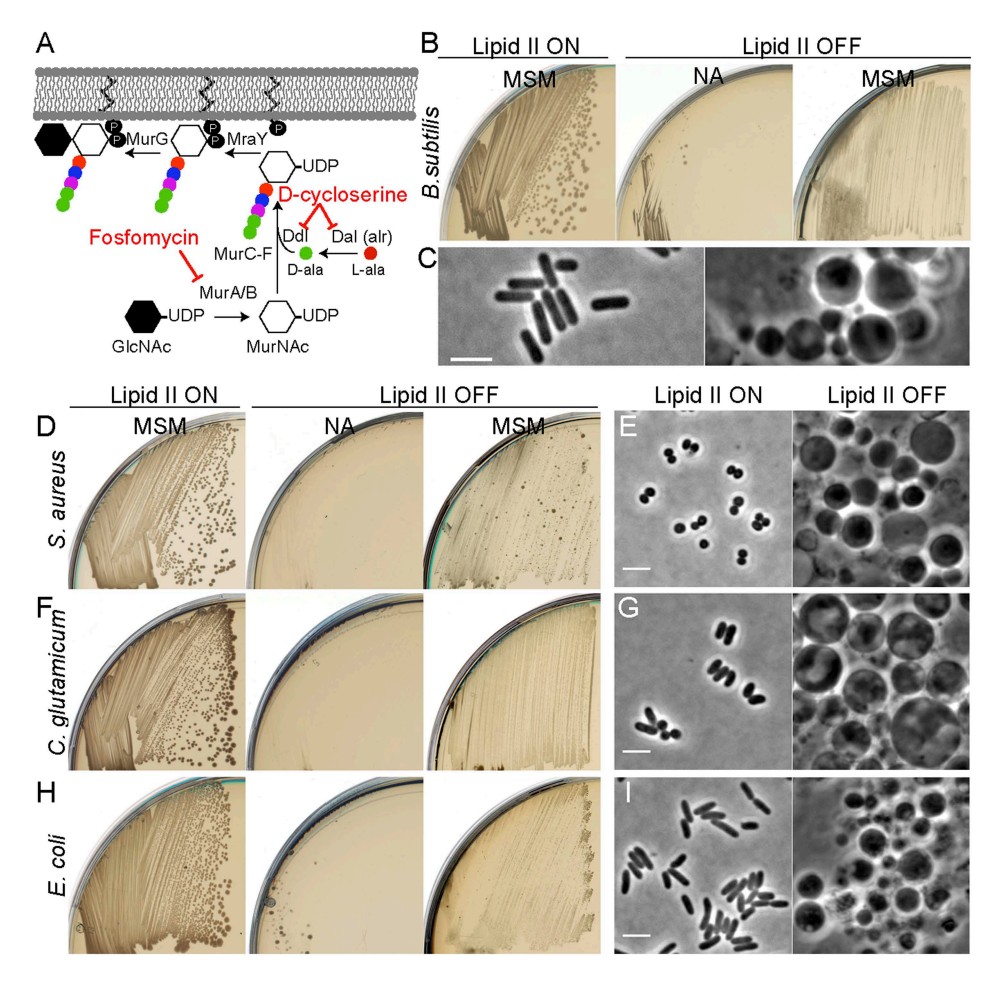

**Figure 1**. Inhibition of PG precursor synthesis induces L-form proliferation in bacteria. (**A**) Schematic model of peptidoglycan (PG) precursor (lipid II) synthesis in bacteria and its inhibition by the antibiotics fosfomycin (FOS) and D-cycloserine (DCS). MurA, inhibited by the antibiotic FOS, and MurB catalyze the transformation of uridine diphosphate-N-acetylglucosamine (UDP-GlcNAc) into UDP-N-acetylmuramic acid (UDP-MurNAc). The racemase Dal and the D-alanine ligase Ddl, both of which are inhibited by the antibiotic DCS, are required to generate D-Ala-D-Ala. This is incorporated into the UDP-MurNAc-pentapeptide, requiring MurC, MurD, MurE, and MurF enzymes. UDP-MurNAc-pentapeptide is transferred to undecaprenyl pyrophosphate by MraY, and the addition of GlcNAc is catalyzed by MurG to form lipid II. (**B**) Growth of *Bacillus subtilis* strain LR2 (*ispA P_xyl-murE-B*) streaked on L-form supporting medium (MSM) or nutrient agar (NA) plates in the presence (lipid II ON) or absence (lipid II OFF) of 0.5% xylose. (**C**) Phase contrast microscopy of *B. subtilis* LR2 cells grown on MSM plates in the presence (left) or absence (right) of 0.5% xylose. (**D–I**) Growth on plates (**D, F, H**) and corresponding phase contrast microscopy (**E, G, I**) of bacterial strains *Staphylococcus aureus* ATCC2913 (**D, E**), *Corynebacterium glutamicum* ATCC13032 (**F, G**), and *Escherichia coli* MG1655 (**H, I**). (**D, F, H**) The different bacterial strains were streaked on MSM or NA plates in the absence (lipid II ON) or presence (lipid II OFF) of the antibiotics FOS (**D, H**) or DCS (**F**). (**E, G, I**) Phase contrast microscopy of the different bacterial cells grown on MSM plates in the absence (left) or presence (right) of the antibiotics FOS (**E, I**) or DCS (**G**). Scale bars, 3 µm.

The following figure supplements are available for figure 1:

**Figure supplement 1**. *Bacillus subtilis* L-form growth requires an additional mutation in a gene such as *ispA*.

**Figure supplement 2**. Bacterial L-forms proliferate on *β*-lactams.

**Figure supplement 3**. Bacterial L-form cell wall reversion.

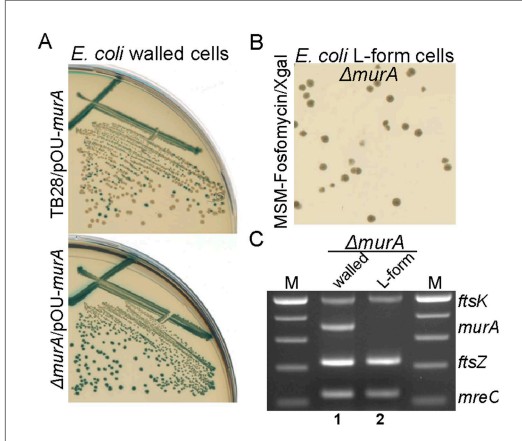

**Figure 2**. *E. coli* L-forms proliferate independently of the peptidoglycan cell wall machinery. (**A**) Growth of the *Escherichia coli* strains TB28 (top) and RM345 (*ΔmurA,* bottom) containing the unstable plasmid pOU82-Amp-*murA* streaked on nutrient agar plates in the presence of X-gal. (**B**) L-form colonies of the *E. coli* strains RM345 (*ΔmurA,* pOU82-Amp-*murA*) on L-form supporting medium (MSM) plates in the presence of fosfomycin (FOS) and X-gal, after several repeated streakings on MSM plates in the presence of FOS. (**C**) Multiplex PCR of the *ftsK, murA, ftsZ,* and *mreC* genes from genomic DNA of the *E. coli* strains RM345 grown in the walled (1) or L-form (2) states, obtained from the strains in panel (**B**). M represents the 100 bp DNA ladder.

(lipid II OFF, NA), growth of *B. subtilis* was restored on osmoprotective NA/supporting medium (MSM) plates (lipid II OFF, MSM), via a switch to an L-form mode of proliferation (*Leaver et al., 2009*; *Mercier et al., 2013*). The gross morphological differences between walled and L-form *B. subtilis* are illustrated in *Figure 1C*. (Note that *B. subtilis* L-form growth requires an additional mutation in a gene such as *ispA* [*Figure 1—figure supplement 1*], for reasons that are not yet clear [*Leaver et al., 2009*].)

We wondered whether similar approaches could be used to elicit L-form growth in other bacteria. To simplify the experiments, we used biochemical inhibitors of the PG precursor pathway, fosfomycin (FOS) or D-cycloserine (DCS), which inhibit the enzymes MurA and Ddl, respectively (*Figure 1A*). We examined three different organisms: two Gram-positive organisms, the Firmicute *Staphylococcus aureus* ATCC29213 and the Actinobacterium *Corynebacterium glutamicum* ATCC13032, and the Gram-negative organism, *E. coli* strain MG1655. In all three cases, we were readily able to generate an L-form transition. *S. aureus* and *E. coli* were both susceptible to FOS at 400 µg/ml. *C. glutamicum* was resistant to FOS but susceptible to DCS at the same concentration. *Figure 1* (D, F, H) shows that the growth of all three strains on NA was inhibited in the presence of the drug (lipid II OFF). However, as observed in *B. subtilis*, growth of all three strains was efficiently restored under osmoprotective conditions (lipid II OFF, MSM). Furthermore, phase contrast microscopy of the three treated cultures (*Figure 1E, G, I*; OFF) revealed the presence of large spheroidal cells strikingly similar to the L-forms of *B. subtilis* (*Figure 1C*) and quite different from the parental walled cells (*Figure 1E, G, I*; ON), consistent with the idea that all three diverse organisms are able to switch to an L-form mode of proliferation on inhibition of the PG precursor pathway. We further showed that L-forms of the three different species could be successively propagated in the presence of high (500 µg/ml) concentrations of β-lactam antibiotics (*Figure 1—figure supplement 2B*) in concentrations that are normally lethal in walled cells (*Figure 1—figure supplement 2A*). Finally, on reactivation of PG precursor synthesis, the three different species readily reverted to their parental walled forms (*Figure 1—figure supplement 3*, *Figure 4—figure supplement 1A–B*), by de novo synthesis of the cell wall sacculus (*Kawai et al., 2014*).

In *B. subtilis*, proliferation in the L-form state renders the normally essential genes of the PG precursor pathway dispensable (*Leaver et al., 2009*; *Mercier et al., 2013*). Thus, to test whether FOS or DCS are sufficient to promote the full switch to an L-form mode of proliferation, we assessed whether the PG precursor pathway genes were essential in the genetically tractable bacterium *E. coli*. We first constructed plasmid pOU82-*murA*, which carried a copy of *murA*⁺ located on an unstable mini-R1 plasmid (*Gerdes et al., 1985*), together with a *lacZ* gene encoding β-galactosidase. In the presence of this plasmid, we were then able to construct a chromosomal deletion of *murA*, which is an essential gene of the PG precursor pathway (*Figure 1A*), giving strain RM345 (*murA::Kn,* pOU82-*murA*). In walled cells, the presence of plasmid pOU82-*murA* was essential for growth of strain RM345 (*Figure 2A*, bottom), as demonstrated by the uniform blue colonies on X-gal, while it was readily lost from the parental TB28 strain (*murA*⁺), giving many white colonies (*Figure 2A*, top). Strikingly, the plasmid was also readily lost from strain RM345 when grown in the putatively L-form state, as indicated by the white colonies (*Figure 2B*). To confirm the specific loss of the *murA* gene, we performed a multiplex PCR (see 'Materials and methods' below) on DNA purified from cells of strain RM345 grown in

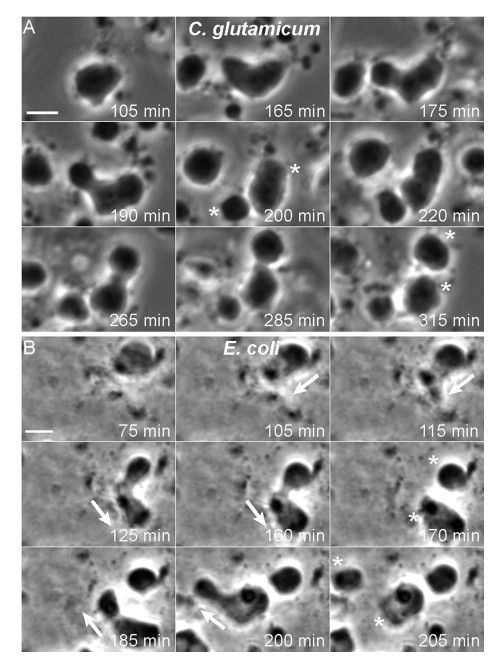

**Figure 3**. Mode of cell division of *E. coli* and *C. glutamicum* L-forms. (**A, B**) *Corynebacterium glutamicum* L-form strain grown in nutrient broth (NB)/L-form supporting medium (MSM) with D-cycloserine (**A**), and *Escherichia coli* L-form strain RM345 (*ΔmurA*) grown on nutrient agar/MSM (**B**), were observed by time lapse phase contrast microscopy. Elapsed time (min) is shown in each panel. Scale bars, 3 μm. Arrows represent the direction of protrusion formation and the asterisks (*) the daughter cells after division. See also **Videos 1–4**.

the presence of the cell wall or as L-forms. As shown in *Figure 2C*, in the walled state, the *murA* gene was readily detected (lane 1) whereas it was not detected in the DNA from a white L-form colony (lane 2).

## *E. coli* and *C. glutamicum* L-forms divide by a classical L-form mechanism

Having created newly growing bacterial L-forms from different bacterial species, we wished to investigate their mode of cell proliferation using time lapse microscopy. For *C. glutamicum*, the L-forms grow readily under various conditions, including liquid media, and we were readily able to capture time lapse sequences that revealed a pattern of proliferative events very similar to those we described previously for *B. subtilis* (*Leaver et al., 2009*; *Mercier et al., 2013*). *Figure 3A* and *Video 1* and *Video 2* show typical time courses. In *Figure 3A*, the central cell underwent repeated shape deformations, with proliferative events generating separate cells after 200 and 315 min (*).

Unfortunately, in the case of *S. aureus*, we have so far been unable to grow them in liquid medium. On appropriate solid medium, although the L-form cultures clearly undergo substantial increases in biomass and the cells have a typical L-form morphology in still images, we have not yet been able to visualize specific division events by time lapse imaging.

Growth of *E. coli* L-forms in liquid media has also been problematic. However, in this case, we have succeeded in capturing suitable time lapse data. *Figure 3B* (and *Video 3* and *Video 4*) show typical examples of an *E. coli ΔmurA* L-form strain grown on solid medium (NA/MSM). Strikingly, the mode of cell proliferation is reminiscent of the Gram-positive *B. subtilis* and *C. glutamicum* L-forms. We observed a repeat cycle of cell deformation and cell protrusion formation at 105–160 min and 185–200 min (arrows), each followed by a spontaneous division generating progeny cells after 170 min and 205 min (*).

It thus appears that the general features of the L-form mode of cell proliferation are conserved between Gram-positive and Gram-negative bacteria.

## Bacterial L-forms divide independently of the normally essential cell division machinery

Cell division of walled bacteria requires the assembly and function of a complex proteinaceous machinery built around the essential tubulin homologue FtsZ (*Adams and Errington, 2009*). We showed previously that in *B. subtilis* L-forms, remarkably the FtsZ protein and probably the whole cell division machinery become dispensable (*Leaver et al., 2009*). We therefore tested the role of the cell division machinery in the newly created bacterial L-forms.

For *E. coli*, we used the method described above for *murA* to construct an *ftsZ* deletion mutant complemented by plasmid pOU82-*ftsZ* (strain RM349, *ftsZ*::Kn, pOU82-*ftsZ*). When RM349 was grown in the walled state, FtsZ appeared essential, as judged by the blue only colonies (*Figure 4A*, bottom). Once again, when induced to grow in the L-form state, FtsZ became dispensable, as characterized by the presence of white colonies on X-gal plates (*Figure 4B*, top left). Multiplex PCR was used to confirm loss of the *ftsZ* gene only in the L-form cell DNA (*Figure 4C*, lanes 1 and 2). Additionally,

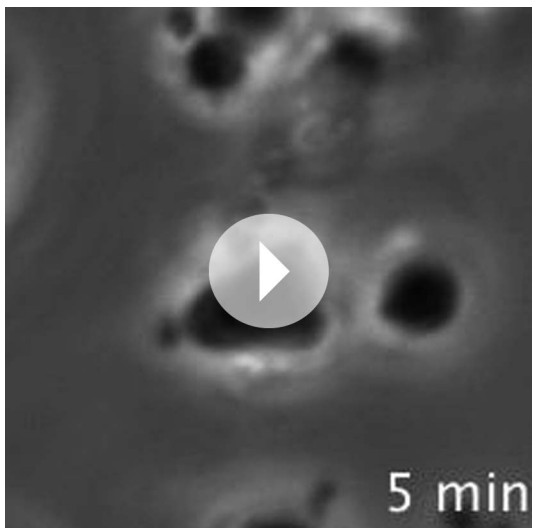

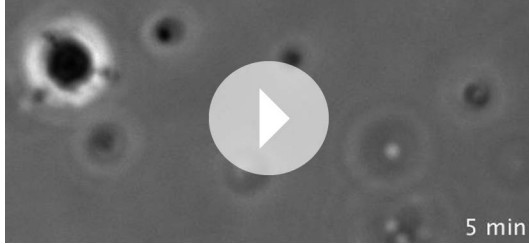

**Video 2**. Time lapse series showing L-form cell growth of *Corynebacterium glutamicum* growing in nutrient broth (NB)/L-form supporting medium (MSM) with D-cycloserine (DCS). Phase contrast images were acquired automatically every 5 min for about 3 hr 30 min. Scale bar, 3 μm.

**Video 1**. Time lapse series showing L-form cell growth of *Corynebacterium glutamicum* growing in nutrient broth (NB)/L-form supporting medium (MSM) with D-cycloserine (DCS), from which the panels in *Figure 3A* were obtained. Phase contrast images were acquired automatically every 5 min for about 5 hr. Scale bar, 3 μm.

using a similar strategy, we showed that both FtsZ and MurA proteins were simultaneously dispensable in L-forms (*Figure 4B*, top right, and *Figure 4C* lanes 3 and 4), as well as another essential cell division protein FtsK (*Figure 3B*, bottom left and 4C lanes 5 and 6), and the cytoskeleton proteins MreBCD (*Figure 4B*, bottom right, and *Figure 4C* lanes 7 and 8).

To examine whether the cell division machinery was essential in *S. aureus*, we took advantage of strain RNpFtsZ_1 (*Pinho and Errington, 2003*) in which the *ftsZ* gene is controlled by an isopropyl β-D-1-thiogalactopyranoside (IPTG) inducible promoter. As expected, this strain was unable to proliferate without inducer in the presence of the cell wall (*Figure 4D*, lipid II ON, −FtsZ). However, when the cells were switched into an L-form mode of proliferation (lipid II OFF), no growth difference was detectable between the presence (+FtsZ) or absence (−FtsZ) of IPTG. To exclude the possibility that the strain picked up a suppressor mutation relieving the dependence of *ftsZ* expression on IPTG, we reverted the L-forms to the parental walled form (by removing FOS) and showed that the cells regained their dependence on IPTG (*Figure 4—figure supplement 1A and 1B*).

As an alternative way to test for dependence on the cell division machinery in *S. aureus*, we used the strain ATCC2913 *ftsZ*[R191P] (*Haydon et al., 2008*), which carries an amino acid substitution in FtsZ that renders the cells dependent on a benzamide antibiotic. Walled cells grow and divide normally in the presence of the antibiotic but the mutant FtsZ protein fails to support division in the absence of benzamide (*Figure 4—figure supplement 1C*, lipid II ON). In accordance with the above results, growth in the absence of benzamide was restored when the cells were switched to the L-form state (*Figure 4—figure supplement 1C*, lipid II OFF), again showing that *S. aureus* L-forms can proliferate independently of FtsZ and hence of the normal cell division machinery.

Construction of conditional mutants of *C. glutamicum* is not as straightforward as for *B. subtilis* or *E. coli*, so, to test the requirement for the cell division machinery in *C. glutamicum* L-forms, we cultured the organism in the walled and L-form states in the presence of cephalexin, a specific inhibitor of the essential cell division protein FtsI (*Pogliano et al., 1997*). As previously shown, cephalexin blocks cell division in normal walled cells (*Valbuena et al., 2006*), leading to a severe growth defect (*Figure 4E*, left and *Figure 4—figure supplement 2*, middle). However, in the L-form mode of proliferation, no growth defect was observed (*Figure 4E*, right and *Figure 4—figure supplement 2*, right), again supporting the idea that L-form proliferation is independent of the normal cell division machinery.

## Regulation of fatty acid synthesis is crucial for proliferation of *E. coli* and *C. glutamicum* L-forms

We recently showed that a minor reduction in fatty acid synthesis, that had no effect on walled cell growth or division, specifically abolished *B. subtilis* L-form proliferation (*Mercier et al., 2013*). To investigate

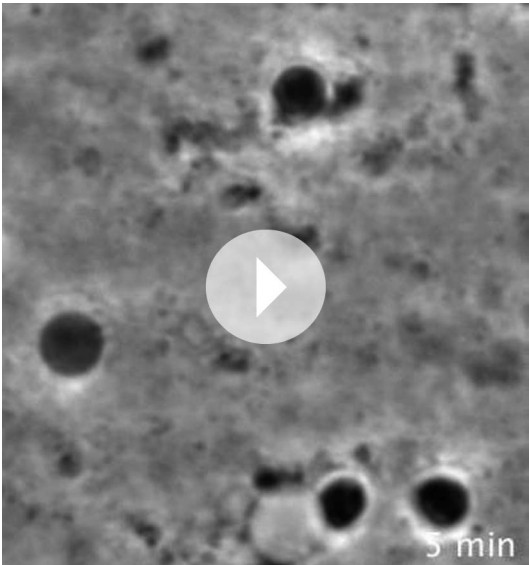

**Video 3**. Time lapse series showing L-form cell growth of *Escherichia coli* strain RM345 (*ΔmurA*) growing in nutrient agar (NA)/L-form supporting medium (MSM) from which the panels in *Figure 3B* were obtained. Phase contrast images were acquired automatically every 5 min for about 4 hr. Scale bar, 3 μm.

**Video 4**. Time lapse series showing L-form cell growth of *Escherichia coli* strain RM345 (*ΔmurA*) growing in nutrient agar (NA)/L-form supporting medium (MSM). Phase contrast images were acquired automatically every 5 min for about 4 hr. Scale bar, 3 μm.

whether similar effects could be observed in the newly characterized bacterial L-forms, we assessed the effects of reductions in the rate of membrane synthesis on L-form proliferation.

In *E. coli*, we used a non-essential fatty acid (FA) synthesis mutant *fabH* previously demonstrated to have a reduced rate of membrane synthesis (*Yao et al., 2012*). As shown in *Figure 5A*, a *fabH* null strain proliferated in the walled state on NA/MSM plates (lipid II ON), while no growth was detected following a switch to the L-form mode of proliferation (lipid II OFF, middle). Importantly, this growth defect was restored by a *fabH⁺* complementing plasmid (*Figure 5A*, left). Similar results were obtained using cerulenin, an antibiotic that inhibits FA synthesis, which specifically inhibited L-form proliferation (*Figure 5—figure supplement 1*). Finally, to demonstrate whether FA synthesis regulation was essential for *E. coli* L-form proliferation, we constructed a strain bearing a double deletion of *murA* and *fabH* (strain RM369, *murA*, fabH::Kn pSK122-*murA*, pOU82-fabH) bearing *murA⁺* on an unstable mini-F plasmid and *fabH⁺* on an unstable mini-R1 plasmid. This strain was grown in both walled and L-form states on NA/MSM plates with no direct selection for the plasmids. After DNA extraction, we assessed the presence of the *murA* and *fabH* genes using multiplex PCR. As expected, in the walled state, the *murA* gene was retained because PG synthesis is essential, but the *fabH* gene was lost, because *E. coli* apparently has a second activity capable of supporting the *fabH* function (*Yao et al., 2012*) (*Figure 5C*, lane 1). Strikingly, in the L-form state, the opposite was observed: *murA* was lost, while *fabH* was retained (*Figure 5C*, lane 2), supporting the idea that a higher rate of FA synthesis is required for proliferation of *E. coli* in the L-form state.

To test whether the rate of FA synthesis is also important for *C. glutamicum* L-form proliferation, we streaked growing walled and L-form cells on NA/MSM plates in the presence of 2 μg/ml of cerulenin. In accordance with the results for *E. coli* (above), partial inhibition of FA synthesis specifically inhibited L-form proliferation (*Figure 5D*, left), with no effect on the walled cells (*Figure 5D*, middle). Time lapse microscopy was used to assess the effects of reduction of FA synthesis on L-form proliferation. As shown in *Figure 5E*, left, and *Video 5*, in the absence of cerulenin, cells grow and divide normally. Strikingly, in the presence of cerulenin, cells continued to grow but shape deformations did not occur, and the cells remained more or less spherical with no detectable division events (*Figure 5E*, right, and *Video 6*).

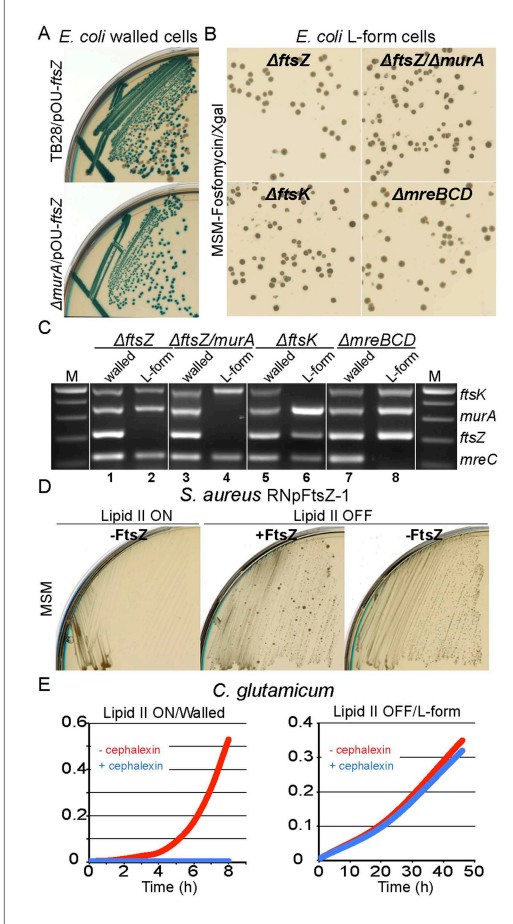

**Figure 4**. Bacterial L-forms proliferate independently of the cell division machinery. (**A**) Growth of the *Escherichia coli* strains TB28 (top) and RM349 (*ΔftsZ*, bottom) containing the unstable plasmid pOU82-Amp-*ftsZ* streaked on nutrient agar plates in the presence of X-gal. (**B**) L-form colonies of the *E. coli* strains RM349 (*ΔftsZ*, pOU82-Amp-*ftsZ*, top left), RM350 (*ΔmurA*, *ΔftsZ*, pOU82-Amp-*ftsZ*, pSK122-Cm-*murA*, top right), RM61 (*ΔftsK*, pSK122-Cm-ftsK, bottom left), and RM359 (*ΔmreBCD*, pHM82-Kn-*mreBCD*) on L-form-supporting medium (MSM) plates in the presence of fosfomycin (FOS) and X-gal, after several repeated streakings on MSM plates in the presence of FOS. (**C**) Multiplex PCR of the *ftsK*, *murA*, *ftsZ*, and *mreC* genes from genomic DNA of the *E. coli* strains RM349 (1, 2), RM350 (3, 4), RM61 (5, 6), and RM359 (7, 8) grown in the walled (1, 3, 5 and 7) or L-form (2, 4, 6 and 8) states obtained from the strains in panel (**B**). M represents the 100 bp DNA ladder. (**D**) Growth of the *Staphylococcus aureus* strain RNpFtsZ-1 (erm-pSPAC-*ftsZ*, **Pinho and Errington, 2003**) streaked on MSM plates in the absence (lipid II ON, left) or presence (lipid II OFF, middle and right) of FOS, with (+FtsZ, middle) or without (−FtsZ, left and right) isopropyl β-D-1-thiogalactopyranoside. (**E**) Growth profiles of *Corynebacterium glutamicum* in MSM with (L-form state; right, lipid II OFF) or without (walled state; left, lipid II OFF)

*Figure 4. Continued on next page*

Thus, as previously described for *B. subtilis* L-forms, regulation of membrane synthesis seems to have a pivotal role in the proliferation of diverse Gram-positive and Gram-negative L-forms.

## Discussion

### Inhibition of PG precursor synthesis induces L-form proliferation in diverse bacteria

Historically, L-forms from diverse bacteria were generated using many different cell wall inhibitors, such as β-lactams, glycopeptides, and lytic enzymes (**Domingue and Woody, 1997**). The wide range of methods used to create L-forms has probably contributed to the heterogeneity in phenotypic properties and has made it difficult to define general properties for L-form bacteria (**Domingue and Woody, 1997**; **Allan et al., 2009**). Included in the range of cells designated L-form 'like' have been cell types in which the PG synthesis machinery remained essential for proliferation (e.g. the *E. coli* cells of **Joseleau-Petit et al., 2007** and **Cambre et al., 2014**). Given that we have now shown that *E. coli* can be converted into a state in which the cell wall precursor pathway can be deleted and cells become completely resistant to β-lactam antibiotics, we suggest that in future the term L-form be restricted to fully wall deficient cells.

We previously showed that for *B. subtilis*, inhibiting an earlier step of the PG precursor pathway efficiently generates proliferating L-forms (**Leaver et al., 2009**; **Dominguez-Cuevas et al., 2012**). Perhaps surprisingly, this approach appears to have been tried only rarely in previous L-form work (**Schmid, 1984**, **1985**). We showed here that inhibition of the PG precursor pathway readily generates L-forms in diverse bacteria of both Gram-positive (*S. aureus* and *C. glutamicum*) and Gram-negative (*E. coli*) varieties. Furthermore, this method generated genuine cell wall-free proliferative bacteria, as their growth was not inhibited by high concentrations of β-lactam antibiotics and, more importantly, essential PG synthesis genes could be deleted (at least for *B. subtilis* and *E. coli*). Finally, as the PG precursor synthesis pathway is almost ubiquitous in bacteria, it is reasonable that this method could be applied to a very wide range of bacteria.

We do not yet understand why PG precursor synthesis inhibition efficiently promotes L-form proliferation from different bacteria. However, we recently uncovered that in *B. subtilis*, PG precursor synthesis inhibition triggers, by an unknown mechanism, induction of an excess of membrane

*Figure 4. Continued*

lipid II ON) D-cycloserine, and in the absence (red) or presence (blue) of cephalexin.

The following figure supplements are available for figure 4:

**Figure supplement 1**. *Staphylococcus aureus* L-forms proliferate in the absence of the cell division machinery.

**Figure supplement 2**. *Corynebacterium glutamicum* L-forms proliferate in the absence of the cell division machinery.

synthesis, a key process for L-form cell division (*Mercier et al., 2013*). Thus, as PG synthesis needs to be coordinated either with membrane synthesis or cell growth, it is plausible that PG precursor synthesis inhibition has general effects on the regulation of membrane synthesis in bacteria.

## General properties of bacterial L-forms

Having created different types of bacterial L-forms, we identified several common and differentiated properties, as summarized in *Table 1*.

(i) Growth conditions. We previously found that *B. subtilis* L-forms can proliferate in both solid and liquid media (*Leaver et al., 2009*). Interestingly, although *C. glutamicum* L-forms shared the ability to proliferate under both conditions, *S. aureus* and *E. coli* L-forms only grew on an agar surface. Another recently characterized L-form, from the bacterium *Listeria monocytogenes,* was also reported to grow only under semi-solid conditions (*Dell'Era et al., 2009*). Thus the ability of L-forms to proliferate under different growth conditions is dependent on as yet unknown inherent properties of each bacterial species.

(ii) Genetic mutations. We previously showed that L-form proliferation in *B. subtilis* requires, in addition to inhibition of PG precursor synthesis, a mutation in a gene such as *ispA* (*Leaver et al., 2009*; *Mercier et al., 2013*). In the absence of such a mutation, no growth is detected on inhibition of PG precursor synthesis (*Figure 1—figure supplement 1*). Interestingly, the different bacteria tested here readily proliferated on inhibition of PG precursor synthesis, strongly suggesting that no *ispA*-like mutation is needed to promote L-form proliferation. Thus it appears again that the requirement for a secondary mutation to promote L-form proliferation will depend on the bacterium tested.

(iii) FtsZ independent cell division. The most remarkable property observed in *B. subtilis* L-forms was a mode of cell division independent of the normally essential protein based machinery (*Leaver et al., 2009*). Remarkably, *S. aureus* and *E. coli* L-forms share the ability to proliferate independently of FtsZ, and although definitive experiments are more difficult to perform in *C. glutamicum*, it appears that they will also share this property. Thus FtsZ independent proliferation is a common trait of bacterial L-forms, presumably reflecting their strange blebbing/tubulation mode of growth. We suggest that the ability to tolerate deletion of essential cell division genes such as *ftsZ* will be a useful operational test for the true L-form state.

(iv) Cell wall reversion. We recently showed that *B. subtilis* L-forms are able to synthesis a cell wall sacculus de novo, followed by reversion to the parental walled form (*Kawai et al., 2014*). Similarly, after reactivating PG precursor synthesis by removal of FOS or DCS, the three different bacterial L-forms tested here could also revert to their parental walled forms, suggesting that the ability to rebuild a cell wall sacculus de novo is also a common property of bacteria.

## A common ancestral mode of cell proliferation in bacterial L-forms

We reported here that *C. glutamicum* and *E. coli* L-forms, at least, appear to proliferate by cell shape deformations followed by spontaneous scission events, in a very similar manner to the process we have described for *B. subtilis* (*Leaver et al., 2009*; *Mercier et al., 2012*, *2013*). Additionally, as previously observed for *B. subtilis*, a minor reduction in FA synthesis prevented the growth of both *C. glutamicum* and *E. coli* L-forms, supporting the idea that all three L-forms divide by a similar mechanism based on an increased ratio of surface area to volume synthesis. Thus it appears that evolutionarily divergent bacteria, with different envelope structures (e.g. Gram-positive and Gram-negative), shape (e.g. rod vs sphere), and different modes of cell wall extension (e.g. lateral and apical) have retained a common primitive mode of proliferation when forced to grow in the absence of a cell wall. Interestingly, this mode of proliferation is strikingly similar to the mode of proliferation of simple vesicle systems independent of protein based machineries (*Hanczyc et al., 2003*; *Budin et al., 2009*; *Terasawa et al., 2012*). Therefore, our results strengthen the idea that the L-form mode of proliferation could have been used by a common ancestor of the bacteria prior to the invention of the cell wall, and are consistent with the notion that invention of the wall was a pivotal moment in the evolutionary divergence of the bacterial lineage (*Errington, 2013*).

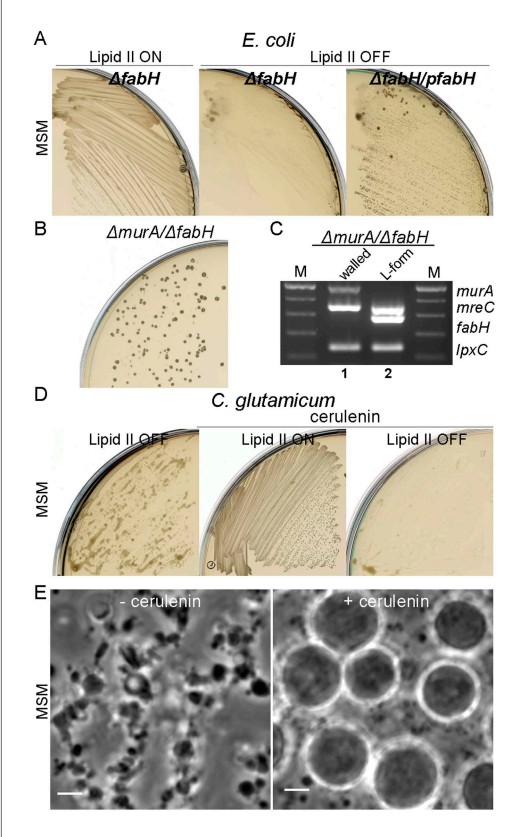

**Figure 5**. Essential role of fatty acid synthesis in L-forms growth of *E. coli* and *C. glutamicum*. (**A**) Growth of *Escherichia coli* strains RM365 (Δ*fabH*) and RM366 (Δ*fabH*, pCA24N-fabH) streaked on L-form supporting medium (MSM) in the absence (lipid II ON) or presence (lipid II OFF) of fosfomycin (FOS). (**B**) L-form colonies of the *E. coli* strain RM369 (Δ*murA*, pSK122-Cm-ftsK, Δ*fabH*, pOU82-Amp-fabH) on MSM plates after several repeated streakings on MSM plates in the presence of FOS. (**C**) Multiplex PCR of the genes, *murA*, *fabH*, and *mreC* on genomic DNA of the *E. coli* strain RM369 grown in the walled (1) or L-form (2) state. Samples obtained from strains in panel (**B**). M represents the 100 bp DNA ladder. (**D**) Growth of *Corynebacterium glutamicum* streaked on MSM in the absence (lipid II ON) or presence (lipid II OFF) of D-cycloserine (DCS), and with (cerulenin) or without (no) 2 μg/ml of cerulenin. (**E**) Typical images of *C. glutamicum* L-forms after 16 hr of growth in MSM with DCS in the absence (−cerulenin) or presence (+cerulenin) of 2 μg/ml of cerulenin. Scale bars, 3 μm. See also *Videos 5 and 6*.

The following figure supplement is available for figure 5:

**Figure supplement 1**. Specific inhibition of *Escherichia coli* L-forms growth by cerulenin.

## Broader implications

We report here a simple and possibly widely generalizable method with which bacteria can be switched to a cell wall-free mode of proliferation. Apart from its apparent importance for understanding an early step in the evolution of life, the simple mechanism of proliferation of L-forms may find application in attempts to design and engineer synthetic self-replicative systems, or minimal cells (*Caspi and Dekker, 2014*). The ability to delete and then restore normally essential genes in L-forms offers a powerful new model system with which to investigate important properties of the cell wall synthesis and cell division machineries, with implications for the discovery and development of novel antibacterials (*Bugg et al., 2011*; *den Blaauwen et al., 2014*).

## Materials and methods

### Bacterial strains and plasmids

The bacterial strains and plasmids constructs used in this study are shown in *Table 2*. DNA manipulations and *E. coli* DH5α transformation were carried out using standard methods (*Sambrook et al., 1989*). The plasmids pOU82-*murA* and pSK122-*murA* contain the operon *yrbA*-*murA*. The plasmids pOU82-*ftsZ* and pOU82-*fabH* contain the *ftsZ* or *fabH* gene, respectively, fused to a constitutive *E. coli* promoter (**ttgaca**gctagc tcagtcctagg**tactgt**gcta) designed by John Anderson (IGEM2006_Berkeley). The *E. coli murA* (RM345) and *ftsZ* (RM349) deletion mutant strains were created using the Lambda Red recombinase system with a derivate of pKD4 as a template (*Datsenko and Wanner, 2000*). Briefly, the strains TB28 containing pOU82-*murA* or pOU82-*ftsZ* and pKD46-sp were transformed by a PCR product containing the kanamycin cassette flanked by 40 nt homology regions, just upstream of the start and downstream of the stop codons, of the genes *murA* or *ftsZ*. Deletions were tested by PCR and backcrossed into fresh TB28 containing pOU82-*murA* or pOU82-*ftsZ* using P1 transduction.

### Growth conditions

The different walled bacterial cells (*B. subtilis*, *E. coli*, *S. aureus*, and *C. glutamicum*) were grown on NA (Oxoid Limited, UK) and in Luria–Bertani broth. Bacterial L-forms were grown in osmoprotective medium composed of 2× MSM media, pH 7 (40 mM MgCl$_2$, 1 M sucrose, and 40 mM maleic acid), mixed 1:1 with 2× nutrient broth (NB, Oxoid) or 2× NBA (NB with 2% agarose). When necessary, antibiotics and supplements were added to media at the following concentrations: FOS 0.4 mg/ml; DCS 0.4 mg/ml; penicillin G 0.5 mg/ml; ampicillin 50 μg/ml or 0.5 mg/ml; chloramphenicol 25 μg/ml;

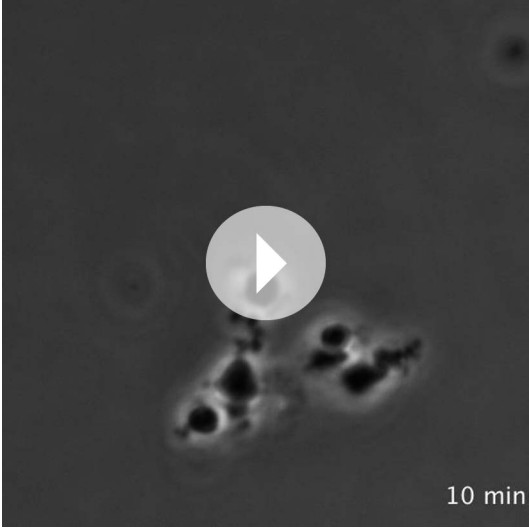

**Video 5**. Time lapse series showing L-form cell growth of *Corynebacterium glutamicum* growing in nutrient broth (NB)/L-form supporting medium (MSM) with D-cycloserine (DCS) in the absence of 2 µg/ml of cerulenin. Phase contrast images were acquired automatically every 5 min for about 16 hr. Scale bar, 3 µm.

**Video 6**. Time lapse series showing L-form cell growth of *Corynebacterium glutamicum* growing in nutrient broth (NB)/L-form supporting medium (MSM) with D-cycloserine (DCS) in the presence of 2 µg/ml of cerulenin. Phase contrast images were acquired automatically every 5 min for about 16 hr. Scale bar, 3 µm.

kanamycin 25 µg/ml; erythromycin 10 µg/ml; cerulenin 2 µg/ml, 10 µg/ml, or 20 µg/ml; xylose 0.5%; IPTG 1 mg/ml; and 1 µg/ml benzamide (FtsZ inhibitor 8J [*Adams et al., 2011*]).

## Multiplex PCR

For multiplex PCR, *E. coli* walled and L-form genomic DNA samples were prepared using a standard phenol–chloroform extraction procedure. The primer couples were designed using MPprimer software (*Shen et al., 2010*), using an open reading frame nucleotide sequence. Standard PCR reaction procedures were applied using GoTaq DNA Polymerase (Promega, Madison, WI) with a melting temperature of 56°C.

## Microscopy and image analysis

For snapshot microscopy, the different bacterial walled and L-form cells were immobilized on microscope slides covered with a thin film of 1% agarose in NB/MSM. The cells were imaged on a

**Table 1.** General properties of bacterial L-forms

| Bacterial L-form strain | *Bacillus subtilis* | *Staphylococcus aureus* | *Corynebacterium glutamicum* | *Escherichia coli* |
|---|---|---|---|---|
| Mode of induction | MurE-B repression | FOS | DCS | FOS |
| Secondary mutation required | Yes | n.d. | n.d. | n.d. |
| Timing of induction | 24 hr | 3 days | 48h | 3 days |
| Growth condition | Solid/liquid | Solid | Solid/liquid | Solid |
| Cell wall reversion | Yes | Yes | Yes | Yes |
| Cell division machinery | Not essential | Not essential | Not essential | Not essential |
| Mode of cell proliferation | Vesicles blebbing, fission, tubulation | n.d. | Vesicles blebbing, fission, tubulation | Vesicles blebbing, fission, tubulation |
| References | *Leaver et al., 2009*, *Mercier et al., 2013*, *Kawai et al., 2014* | | | |

DCS: D-cycloserine; FOS: fosfomycin; n.d.: not determined.

**Table 2.** Bacterial strains and plasmids used in this study

| Strain | Relevant genotype | Reference |
|---|---|---|
| **Bacillus subtilis** | | |
| Bs115 | 168CA *ΩspoVD::cat* *P_{xyl}-murE ΩamyE::(tet xylR)* | (*Leaver et al., 2009*) |
| LR2 | Bs115 *xseB** (frameshift 22T > −)[a] | (*Mercier et al., 2013*) |
| **Escherichia coli** | | |
| TB28 | MG1655 *ΔlacIZYA* | (*Bernhardt and de Boer, 2003*) |
| ND101 | *fstK::Kn* pSC101-fstK | F-X Barre Lab unpublished |
| RM61 | TB28 *ΔftsK::kan* pSK122-ftsK | This study |
| RM345 | TB28 *ΔmurA::kan* pOU82-murA | This study |
| RM349 | TB28 *ΔftsZ::kan* pOU82-ftsZ | This study |
| RM350 | TB28 *ΔftsZ* pOU82-ftsZ *ΔmurA::kan* pSK122-murA | This study |
| RM359 | TB28 *ΔmreBCD::cat* pTK549 | (*Kruse et al., 2005*) |
| RM365 | TB28 *ΔfabH::kan* | (*Baba et al., 2006*) |
| RM366 | RM365 pCA24N-*fabH* | This study |
| RM369 | TB28 *ΔfabH* pOU82-fabH *ΔmurA::kan* pSK122-murA | This study |
| **Staphylococcus aureus** | | |
| WT | ATCC29213 | Laboratory collection |
| *S. aureus ftsZ^{R191P}* | ATCC29213 *ftsZ^{R191P}* | (*Haydon et al., 2008*) |
| RNpFtsZ-1 | RN4220 *P_{spac}-ftsZ erm* | (*Pinho and Errington, 2003*) |
| **Corynebacterium glutamicum** | | |
| WT | ATCC13032 | Laboratory collection |

| Plasmid | Relevant genotype | Reference/origin |
|---|---|---|
| **Escherichia coli** | | |
| pCA24N-*fabH* | *lacIq pT5-lac-fabH cat* | (*Kitagawa et al., 2005*) |
| pOU82 | R1-replicon, *bla lacZYA* | (*Gerdes et al., 1985*) |
| pOU82-*murA* | R1-replicon, *bla lacZYA murA* | This study |
| pOU82-*ftsZ* | R1-replicon, *bla lacZYA ftsZ* | This study |
| pOU82-*fabH* | R1-replicon, *bla lacZYA fabH* | This study |
| pTK549 | R1-replicon, kan *P_{mre}-mreBCD* | (*Kruse et al., 2005*) |
| pSK112 | F-replicon, *cat lacZYA* | F-X Barre Lab unpublished |
| pSK112-*ftsK* | F-replicon, *cat lacZYA ftsK* | F-X Barre Lab unpublished |
| pSK112-*murA* | F-replicon, *cat lacZYA murA* | This study |

bla: b-lactamase; cat: chloramphenicol; erm: erythromycin; lacZ: *β*-galactosidase; kan: kanamycin; tet: tetracyclin.

Zeiss Axiovert 200M microscope controlled by Metamorph 6 (Molecular Devices, Sunnyvale, CA) with a Zeiss ×100 Plan-Neofluar oil immersion objective.

For time lapse microscopy, *C. glutamicum* L-form cells were imaged in ibiTreat adherent, 35 mm sterile glass bottom microwell dishes (ibidi GmbH, Munich, Germany). Briefly, an 0.1 ml sample of exponential phase *C. glutamicum* L-form was added to 0.5 ml of fresh NB/MSM and incubated in the microwell dish for 15 min. The cells were washed three times with NB/MSM, and 0.5 ml of fresh NB/MSM with DCS was finally added. For *E. coli*, L-form cells were immobilized on microscope slides covered with a thin film of 1% agarose in NB/MSM with FOS. The cells were imaged on a DeltaVision RT microscope (Applied Precision, Issaquah, WA) controlled by softWoRx (Applied Precision) with a Zeiss ×100 apo

fluor oil immersion lens. A Weather Station environmental chamber (Precision Control) regulated the temperature of the stage.

Pictures and videos were prepared for publication using ImageJ (http://rsb.info.nih.gov/ij) and Adobe Photoshop.

## Acknowledgements

We thank Kenn Gerdes for the gift of various *E. coli* strains and plasmids, François-Xavier Barre for the gift of the strain ND101 and the plasmids pSK122 and pSK123, Heath Murray for the suggestion of the multiplex PCR assay and critical reading of the manuscript, and Waldemar Vollmer for critical reading of the manuscript. This work was funded by a European Research Council Advanced Investigator grant (# 250363; 'OPAL') to JE.

## Additional information

### Funding

| Funder | Grant reference number | Author |
| --- | --- | --- |
| European Research Council | 250363; OPAL | Romain Mercier, Yoshikazu Kawai, Jeff Errington |

The funder had no role in study design, data collection and interpretation, or the decision to submit the work for publication.

### Author contributions

RM, Conception and design, Acquisition of data, Analysis and interpretation of data, Drafting or revising the article; YK, Analysis and interpretation of data, Drafting or revising the article; JE, Conception and design, Drafting or revising the article

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
