## [Decision Letter]

Thank you for sending your work entitled “General principles of L-form formation and proliferation in bacteria” for consideration at *eLife.* Your article has been favorably evaluated by Richard Losick (Senior editor), Roberto Kolter (Reviewing editor), and 2 reviewers, one of whom, Gilles van Wezel, has agreed to reveal his identity.

This manuscript presents a significant advance in the evolving bacterial “L-form” story that is being pursued by the Errington group. Earlier molecular work was focused on L-forms in *Bacillus* and thus how general the phenomenon would be was not apparent. Here, studies have been extended to several other bacteria, *E. coli*, *S. aureus* and the actinobacterium *Corynebacterium*. As a consequence, several common principles of L-form proliferation are established, e.g. reduced or blocked peptidoglycan synthesis and excessive membrane synthesis. These commonalities are exciting in that they demonstrate that the same principles apply in completely different bacteria, i.e. Gram (+) and Gram (-), and those that grow by lateral wall extension (*Bacillus*, *E. coli*) as well as at the apex (*Corynebacterium*). Also, the work is very well carried out, and wonderful movies are provided of L-form proliferation. Overall the manuscript is in excellent shape. However, we all felt that the results should be discussed from an evolutionary perspective. These are very important findings that significantly reinforce the idea that L-form growth represents a primitive mode of proliferation that was likely used by the last common ancestor of bacteria and this should be discussed. The authors can easily accomplish this with some text revision. In addition, we felt that the authors should tone down the claims that this work provides major new molecular insights. The key contribution is that of establishing the generality of L-form proliferation.

Points to address:

1) The work is important from the view of evolution, as it shows that common principles apply and also that this may have been a primitive form of life. The evolutionary aspects could/should therefore be emphasized more. Specifically, the observation that both Gram+ and Gram- bacteria share a common 'primordial' proliferation mechanism but have subsequently diverged and evolved such different cell envelope structures, is an aspect that deserves more attention. This would be useful as the main conclusions from this manuscript are evolution-oriented. What happens with the outer membrane, e.g. when *E. coli* cells are turned into L-forms? And how is the outer membrane structure restored once L-forms are allowed to resynthesize PG? Although the underlying mechanism might be beyond the scope of this story, it would at least be worth to speculate about the major differences between these different prokaryotes. It is also important to reflect on how the cells (L forms) protect themselves as it is hard to believe the membrane is the only barrier with the outside world. Maybe comparison to protoplasts is interesting, though admittedly highly speculative.

2) It is prudent to avoid the implicit suggestion that this work provides novel insights into L-form proliferation at the molecular level. The authors state on that previous work has been largely “descriptive and anecdotal”. This current paper does not, however, provide new molecular insights over e.g. Mercier 2012 and Leaver 2009, but rather extends the knowledge of Bacillus to other bacteria. The fact that the cell division machinery (Leaver 2009) or MreB (Mercier 2012) are not required for L-form growth was published in previous work, as was the correlation between PG and membrane synthesis (Mercier 2013) and the mode of proliferation. Therefore, the main advance of this work lies in the direction of evolution, showing that the discoveries made in Bacillus are in fact common principles shared by different bacteria.

3) The authors mentioned that the L-forms of the various organisms can revert back to normal walled cell growth modes, but the data was not shown. Because this is an important demonstration, I would like to see the data included, and potentially some (at least rough) estimate of how efficient the reversion process is in the different organisms.

4) Part of the impact of this paper is the establishment of genetic methods to make L-forms in other organisms for further study. Therefore, better descriptions of strain construction methods are needed so that other groups can repeat the constructions. Please add information on the exact nature of each deletion used (beginning and end point of each deletion), how deletion alleles were introduced, what were the conditions for selecting for the deletion alleles, etc.

---

## [Author Response]

*1) The work is important from the view of evolution, as it shows that common principles apply and also that this may have been a primitive form of life. The evolutionary aspects could/should therefore be emphasized more. Specifically, the observation that both Gram+ and Gram- bacteria share a common 'primordial' proliferation mechanism but have subsequently diverged and evolved such different cell envelope structures, is an aspect that deserves more attention. This would be useful as the main conclusions from this manuscript are evolution-oriented. What happens with the outer membrane, e.g. when* E. coli *cells are turned into L-forms? And how is the outer membrane structure restored once L-forms are allowed to resynthesize PG? Although the underlying mechanism might be beyond the scope of this story, it would at least be worth to speculate about the major differences between these different prokaryotes. It is also important to reflect on how the cells (L forms) protect themselves as it is hard to believe the membrane is the only barrier with the outside world. Maybe comparison to protoplasts is interesting, though admittedly highly speculative*.

We have improved the Discussion to strength the idea that bacteria with different envelope structure and shape have a common way to proliferate in the absence of a cell wall, suggesting that L-forms may demonstrate how primitive cells may have proliferated.

We have evidence that the outer membrane is retained in L-forms, an area of research, we are currently developing. We felt it premature at this stage to speculate in detail on the consequences of loss of the PG layer on OM function, or on the question of restoration upon PG resynthesis.

*2) It is prudent to avoid the implicit suggestion that this work provides novel insights into L-form proliferation at the molecular level. The authors state on that previous work has been largely “descriptive and anecdotal”. This current paper does not, however, provide new molecular insights over e.g. Mercier 2012 and Leaver 2009, but rather extends the knowledge of Bacillus to other bacteria. The fact that the cell division machinery (Leaver 2009) or MreB (Mercier 2012) are not required for L-form growth was published in previous work, as was the correlation between PG and membrane synthesis (Mercier 2013) and the mode of proliferation. Therefore, the main advance of this work lies in the direction of evolution, showing that the discoveries made in Bacillus are in fact common principles shared by different bacteria*.

As suggested by the referee, we have modified the manuscript to tone down the suggestion that this work provides major new molecular insights.

*3) The authors mentioned that the L-forms of the various organisms can revert back to normal walled cell growth modes, but the data was not shown. Because this is an important demonstration, I would like to see the data included, and potentially some (at least rough) estimate of how efficient the reversion process is in the different organisms*.

We have now included a supplemental figure (Figure 1—figure supplement 3) showing the ability of *E. coli* and *C. glutamicum* L-forms to revert. The ability of *S. aureus* was already present in the Figure 4—figure supplement 1.

However, we have not yet been able to grow *E. coli* or *S. aureus* L-forms in liquid media, so we cannot at this stage quantify the efficiency of reversion.

*4) Part of the impact of this paper is the establishment of genetic methods to make L-forms in other organisms for further study. Therefore, better descriptions of strain construction methods are needed so that other groups can repeat the constructions. Please add information on the exact nature of each deletion used (beginning and end point of each deletion), how deletion alleles were introduced, what were the conditions for selecting for the deletion alleles, etc*.

A better description of the plasmid and strain constructions has been added in the experimental procedure for other group to repeat. Furthermore, the different published plasmids and strains will be of course available upon request.